# Association between practice coding of chronic kidney disease (CKD) in primary care and subsequent hospitalisations and death: a cohort analysis using national audit data

Faye Cleary ![ORCID],[1] Lois Kim,[2] David Prieto-Merino,[1] David Wheeler,[3] Retha Steenkamp,[4] Richard Fluck,[5] David Adlam,[6] Spiros Denaxas,[7,8] Kathryn Griffith,[9] Fiona Loud,[10] Sally Hull ![ORCID],[11] Ben Caplin ![ORCID],[3] Dorothea Nitsch[1]

For numbered affiliations see end of article.

**Correspondence to**
Faye Cleary;
faye.cleary@lshtm.ac.uk

## ABSTRACT

**Objective** To examine the association between practice percentage coding of chronic kidney disease (CKD) in primary care with risk of subsequent hospitalisations and death.

**Design** Retrospective cohort study using linked electronic healthcare records.

**Setting** 637 general practitioner (GP) practices in England.

**Participants** 167 208 patients with CKD stages 3–5 identified by 2 measures of estimated glomerular filtration rate <60 mL/min/1.73 m², separated by at least 90 days, excluding those with coded initiation of renal replacement therapy.

**Main outcome measures** Hospitalisations with cardiovascular (CV) events, heart failure (HF), acute kidney injury (AKI) and all-cause mortality

**Results** Participants were followed for (median) 3.8 years for hospital outcomes and 4.3 years for deaths. Rates of hospitalisations with CV events and HF were lower in practices with higher percentage CKD coding. Trends of a small reduction in AKI but no substantial change in rate of deaths were also observed as CKD coding increased. Compared with patients in the median performing practice (74% coded), patients in practices coding 55% of CKD cases had a higher rate of CV hospitalisations (HR 1.061 (95% CI 1.015 to 1.109)) and HF hospitalisations (HR 1.097 (95% CI 1.013 to 1.187)) and patients in practices coding 88% of CKD cases had a reduced rate of CV hospitalisations (HR 0.957 (95% CI 0.920 to 0.996)) and HF hospitalisations (HR 0.918 (95% CI 0.855 to 0.985)). We estimate that 9.0% of CV hospitalisations and 16.0% of HF hospitalisations could be prevented by improving practice CKD coding from 55% to 88%. Prescription of antihypertensives was the most dominant predictor of a reduction in hospitalisation rates for patients with CKD, followed by albuminuria testing and use of statins.

**Conclusions** Higher levels of CKD coding by GP practices were associated with lower rates of CV and HF events, which may be driven by increased use of antihypertensives and regular albuminuria testing, although residual confounding cannot be ruled out.

## STRENGTHS AND LIMITATIONS OF THIS STUDY

⇒ A large database of 167 208 patients with biochemical evidence of chronic kidney disease (CKD) was used for analysis, covering 637 general practitioner practices in England who volunteered to participate in an audit of care and which were representative of the general population in terms of age and sex.

⇒ Risk of confounding due to patient characteristics is reduced by studying the association between practice level (rather than patient level) CKD coding and patient-level outcomes, where practice casemix is not expected to differ with practice coding rates.

⇒ Practice behaviours associated with CKD coding performance that are not believed to occur as a consequence of CKD coding may confound associations but were adjusted for as far as possible.

⇒ Average duration of follow-up between assessment of practice coding performance and end of data collection for outcomes was limited to approximately 4 years; longer term effects of CKD coding may, therefore, not be captured in this study.

## INTRODUCTION

Chronic kidney disease (CKD) is a growing public health problem.[1–3] Consequences of CKD include cardiovascular (CV) morbidity, acute kidney injury (AKI) and premature mortality, with increasing risks as disease progresses.[4] The burden of CKD and associated healthcare costs are increasing,[5 6] yet recognition of the disease in routine practice is often poor and varies between healthcare providers.[7 8] This may lead to delayed intervention and worsen prognosis in many patients with CKD.

In the UK, computer systems used by general practitioner (GP) practices allow electronic coding of patient clinical information, enabling consistent and specific recording

and ease of access to coded data.[9] The National Institute for Health and Care Excellence provides recommendations for regular renal function testing and CKD management in primary care,[10] and the Quality and Outcomes Framework provides financial incentives for GPs to maintain a practice register of patients with CKD stages 3–5.[11]

The National Chronic Kidney Disease Audit (NCKDA) conducted in England and Wales in 2015–2016 found that approximately 30% of biochemically confirmed CKD stages 3–5 cases were not given an appropriate CKD code in primary care electronic healthcare records (EHRs).[8] Among patients with biochemical evidence of CKD stages 3–5, those registered with a CKD code had significantly lower rates of CV hospitalisations, AKI and mortality than those without CKD codes.[12] However, analyses only adjusted for age, sex and coded diabetes, hypertension and CV disease (CVD), due to limited data availability. Therefore, the audit report cautioned that a causal association cannot be established for the reported benefits of CKD coding, as results will be affected by confounding by patient health seeking behaviours and unmeasured morbidities. Some have cautioned against overdiagnosis of CKD that may fail to benefit the overall health of the population,[13] and more research is needed to study the benefits of CKD coding.

In attempt to overcome the issue of unmeasured confounding experienced in original analyses of the NCKDA,[12] we set out to examine the association between completeness of CKD coding of the GP practice at which patients were registered and individual adverse outcomes known to be associated with CKD. We hypothesised that practice CKD coding performance would not be associated with individual patient characteristics within practices (casemix), thereby removing some potential confounding, and, if appropriately adjusted for practice behaviours, analysis would provide stronger evidence for a causal effect of coding of CKD on outcomes. Additional aims were to explore the role of practice behaviours that may occur as a result of CKD coding in reducing risk of adverse outcomes. Evidence that higher levels of practice CKD coding improved patient outcomes would have important ramifications for GPs and may influence policy to improve recognition of CKD in primary care.

## METHODS
### Study design
We carried out a retrospective cohort study using routinely collected EHRs.

### Data sources
The NCKDA database holds selected data from the EHRs of 695 GP practices in England and was used to identify a cohort of patients with CKD for analysis and to define exposure variables. Data extraction ranged from March 2015 to July 2016 as practices were gradually recruited into the audit, with the majority of practices recruited by July 2015. In brief, the audit is a snapshot of care at the time point of audit data extraction. Details of the audit and data collection strategy are specified elsewhere.[8] The NCKDA database was linked to Hospital Episode Statistics (HES) holding information on all hospital admissions in England, and Office for National Statistics mortality data, to followup patients for adverse outcomes. Linkage was carried out by National Health Service (NHS) Digital using NHS number, and hospital record information with pseudoanonymised linkage IDs were provided for analysis.

### Study population
The analysis cohort included all adult patients in the NCKDA database extracted from eligible GP practices in England and with biochemical evidence of CKD (hereafter referred to simply as 'CKD' and/or 'confirmed CKD'), defined as at least two records of estimated glomerular filtration rate (eGFR) $<60\,\text{mL/min/m}^2$ separated by at least 90 days, with the most recent measure recorded within the last 2 years prior to data extraction and excluding any patients with coded initiation of renal replacement therapy.

### Primary exposure
At the time of data extractions (2015–2016), Read codes were used to electronically record patient findings in GP computer systems.[14] Variables defined based on eGFR use the isotopedilution mass spectrometry calibrated Modification of Diet in Renal Disease study equation, the standard GFR-estimating equation in use during the period of data collection.

Practice CKD coding performance was characterised as the percentage of patients with CKD in a practice with a CKD stages 3–5 Read code, hereafter referred to as practice CKD coding (performance) or percent coded CKD. Practice CKD coding performance was defined at practice data extraction, which marked the index date for commencement of follow-up for outcomes.

Practices with fewer than 50 total CKD cases were excluded from analysis due to anticipated excess noise in measurement of the primary exposure.

### Outcomes
Four outcomes of interest were studied: (1) hospitalisation for CV events, (2) hospitalisation for heart failure (HF), (3) hospitalisation with AKI and (4) all-cause mortality, defined in online supplemental table 1. Follow-up began at the time of practice data extraction and was capped at 1 March 2019 for hospital outcomes and 1 September 2019 for deaths.

### Practice features
Features of a practice that may confound the association between practice CKD coding and patient adverse outcomes were defined and categorised as: those reflecting overall practice risk profile; practice testing behaviours for CKD; and characteristics of the identified practice CKD population. Practice features that may improve CKD outcomes, some of which may lie on the

causal pathway between practice CKD coding and patient adverse outcomes, were also defined. Practice percentage variables were defined by summing the number of patients meeting relevant risk factor criteria in each practice and dividing by relevant practice denominators. Practice list size data were used to determine size of the adult population in each practice.

Patient-level risk factors identified in NCKDA data (and used to calculate practice percentages) were defined based on presence of any relevant Read code in the EHR prior to data extraction and included:

► Diabetes—any diabetes code not superseded by a diabetes resolved code
► Hypertension—any hypertension code
► CVD—any CVD code
► CKD stages 3b-5—confirmed CKD and latest eGFR <45
► Statin use—any statin prescription or contraindication code
► Antihypertensive use—any prescription or contraindication code for angiotensin-converting enzyme inhibitors (ACEi) or angiotensin-II receptor blockers (ARBs)
► Blood pressure (BP) targets met—last BP within target range within last year before data extraction: systolic blood pressure (SBP) <130 mm Hg and diastolic blood pressure (DBP) <80 mm Hg in those with diabetes or last urine albumin to creatinine ratio (ACR) ≥70 mg/mmol or last protein to creatinine ratio (PCR) ≥100 mg/mmol; SBP <140 mm Hg and DBP <90 mm Hg in all other patients
► Influenza vaccination—any influenza vaccination code in the last year
► Pneumococcal vaccination—any pneumococcal vaccination code in the last 5 years

Additional patient-level risk factors not available in NCKDA data were defined using HES data and included:

► Recent chronic obstructive pulmonary disease (COPD) admission—admission in last 3 years prior to (NCKDA) practice data extraction with a COPD ICD-10 code (J44) as primary diagnosis
► Recent cancer admission—admission in last 3 years prior to (NCKDA) practice data extraction with a cancer ICD-10 code (C00-C97, excluding non-melanoma skin cancers C44) as primary diagnosis

### Practice characteristics reflecting overall practice risk profile

GPs' awareness on how to identify CKD may depend on the overall burden of conditions associated with CKD in their practice. Practice prevalence of diabetes, hypertension and CVD were determined by summing the number of adult patients meeting patient-level comorbidity definitions, with adult population size as denominator. Practice list size data stratified by age and sex were used to determine mean practice age and percent of adults that were male. Practice deprivation was summarised using the median rank of the Index of Multiple Deprivation (IMD) score among all patients extracted from the GP practice (which was limited to patients with CKD risk factors, creatinine assessments or renal codes).

### Practice testing behaviours

Testing behaviours which may impact the patient vintage (ie, the underlying duration of CKD at detection by the GP) and the types of patients with CKD selected for analysis were also defined within practices, including percentage of diabetes patients with a GFR test in the last year, percentage of patients with CKD with a GFR test in the last year and percentage of the practice adult population with confirmed CKD.

### Practice characteristics of the detected CKD population

Underlying practice morbidity and testing behaviours may impact the types of patients with CKD detected and therefore included in analysis. Percent CKD stages 3b-5, percent recent COPD admission and percent recent cancer admission were defined using patient-level risk factor definitions, with number of total CKD cases as denominator.

### Practice behaviours that may improve CKD outcomes

Practice behaviours expected to be related to CKD coding, some of which may be on the causal pathway from CKD to improved outcomes were defined as: percent usage of ACEi/ARBs in hypertension, percent usage of statins in diabetes, percent usage of statins in CVD, percent meeting BP target in last year in CKD, percent ACR/PCR test in last year in CKD, percent influenza vaccination in last year in CKD and percent pneumococcus vaccination in past 5 years in CKD stages 4–5. Practice behaviour variables were dichotomised at the median value.

### Statistical methods

Baseline characteristics of the study population were summarised by sextile of practice percent coded CKD to determine balance in patient characteristics according to primary exposure. Practice characteristics were also summarised by sextile of practice CKD coding to identify any associations with other practice characteristics.

### Main Cox regression analyses

Cox proportional hazards regression (time to first event) was used to evaluate the association between practice CKD coding and each of the four patient outcomes. Hospitalisation outcomes were censored for death. A 5 knot spline was used for the primary exposure, providing flexibility to demonstrate the nature of association between practice coding and outcomes across the spectrum of practice CKD coding performance, without overfitting. The following adjustments for practice characteristics (included as continuous covariates) were carried out sequentially:

Model 1: adjusted for practice characteristics reflecting overall practice risk profile (primary analysis, planned a priori)

Model 2: adjusted for practice characteristics reflecting overall practice risk profile, practice testing behaviours

and practice characteristics of the detected CKD population (secondary analysis, data driven).

Adjusted HR curves for outcomes with 95% CIs were plotted across the spectrum of practice CKD coding, compared with average (median) practice CKD coding. Attributable fractions for the number of (first) events preventable by the median follow-up time among patients with CKD in practices at lower coding levels (17th percentile, bottom of sextile 2) if such practices instead coded at higher coding levels (83rd percentile, top of sextile 5) were estimated under assumption of causality following adjusted Cox regression (model 2), detailed in online supplemental information 1.

Additional analyses with a single linear continuous covariate for percent coded CKD were carried out after visual inspection of an approximately linear relationship for some model 2HR curves, allowing a more convenient clinical interpretation. These models were restricted to sextiles 2–5 of practice coding only (representing the 67% of most averagely performing practices) where linear trends were most apparent. Descriptive likelihood ratio tests were used to assess improvement in model fit using spline terms vs a single linear covariate.

### Subgroup analyses

Subgroup analyses were carried out by diabetes status and by CKD severity (stage 3a, stage 3b–5). Practice percent coded CKD was recalculated within each subgroup for analysis, since coding behaviour differed substantially between subgroups.

### Analyses of practice behaviours that may improve CKD outcomes

Further Cox regression analyses aimed to identify practice behaviours associated with improvements in all four patient outcomes, with adjustment for all (model 2) confounders as well as practice behaviours that may improve CKD outcomes. Practice CKD coding covariates were excluded from analysis to identify practice factors most predictive of outcomes, regardless of CKD coding performance and not conditional on CKD coding.

### Patient and public involvement

Kidney Care UK supported the research questions, grant applications and related record linkage applications of the NCKDA. After NCKDA discontinuation, Kidney Care UK helped with ethics and section 251 permissions to maintain database access for research purposes. A patient representative (Fiona Loud) was involved in the NCKDA from inception, is a co-author and critically reviewed content of this paper.

## RESULTS

### Data completeness

Of 695 practices in England captured in the NCKDA database, 637 practices (92%) met criteria for analysis (at least 50 CKD cases), covering 99% of all patients with CKD from the original database (n=167 208) (figure 1). CKD coding rates did not differ after excluding ineligible practices, overall or by subgroup, but sample sizes were smaller in some subgroups after excluding practices with fewer than 50 CKD cases (online supplemental table 2).

### Patient characteristics

Study population characteristics were generally well balanced between sextiles of practice CKD coding (table 1). There were trends of slightly higher rates of diabetes, hypertension and CVD coding and lower eGFR in patients in the highest coding practices (sextile 6) and slightly lower rates of comorbidities and higher eGFR in the lowest coding practices (sextile 1), indicating potential differences in either true underlying morbidity or risk factor coding in patients in practices performing at the extremes. Median month of data extraction and resulting follow-up duration were well balanced between sextiles, suggesting good balance in seasonal coverage.

### Practice characteristics

Median practice percent coded CKD was 73.9% in the overall CKD population. It was higher in CKD stages 3b-5 (87.9%) than CKD stage 3a (64.8%), and higher in

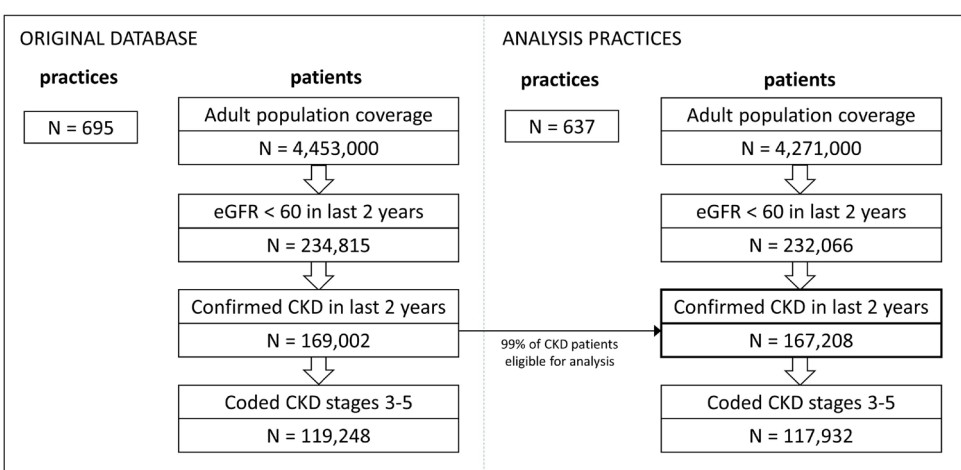

**Figure 1** Flow chart of selection of study population (confirmed CKD in last 2 years). CKD, chronic kidney disease.

**Table 1** Study population characteristics by practice coding sextile

| Practice coding sextile (S) | S1 | S2 | S3 | S4 | S5 | S6 |
|---|---|---|---|---|---|---|
| Practice percent coded CKD | <54.8% | 54.8%–65.5% | 65.5%–73.9% | 73.9%–80.7% | 80.7%–87.5% | ≥87.5% |
| No of patients with CKD | N=29 389 | N=30 360 | N=28 248 | N=28 824 | N=26 731 | N=23 656 |
| Coded CKD | 13 105 (44.6%) | 18 505 (61.0%) | 19 744 (69.9%) | 22 326 (77.5%) | 22 524 (84.3%) | 21 728 (91.8%) |
| Age, mean (SD) | 76.7 (11.2) | 77.1 (11.0) | 77.0 (10.9) | 77.2 (10.8) | 77.1 (10.7) | 77.5 (10.7) |
| Male | 11 686 (39.8%) | 12 240 (40.3%) | 11 464 (40.6%) | 11 709 (40.6%) | 10 664 (39.9%) | 9493 (40.1%) |
| Diabetes | 7013 (23.9%) | 7657 (25.2%) | 6887 (24.4%) | 7288 (25.3%) | 6544 (24.5%) | 6148 (26.0%) |
| Hypertension | 19 475 (66.3%) | 20 749 (68.3%) | 19 300 (68.3%) | 19 798 (68.7%) | 18 860 (70.6%) | 17 207 (72.7%) |
| CVD | 9324 (31.7%) | 9945 (32.8%) | 9087 (32.2%) | 9642 (33.5%) | 8819 (33.0%) | 8086 (34.2%) |
| Biochemical CKD stage | | | | | | |
| 3a | 18 927 (64.4%) | 19 201 (63.2%) | 17 938 (63.5%) | 18 151 (63.0%) | 16 858 (63.1%) | 14 774 (62.5%) |
| 3b | 8384 (28.5%) | 8938 (29.4%) | 8254 (29.2%) | 8576 (29.8%) | 7847 (29.4%) | 7089 (30.0%) |
| 4 | 1838 (6.3%) | 1937 (6.4%) | 1812 (6.4%) | 1831 (6.4%) | 1796 (6.7%) | 1572 (6.6%) |
| 5 | 240 (0.8%) | 284 (0.9%) | 244 (0.9%) | 266 (0.9%) | 230 (0.9%) | 221 (0.9%) |
| Time since last GFR measure (months), median (IQR) | 5.0 (2.0–9.4) | 4.5 (1.8–8.7) | 4.6 (1.9–8.6) | 4.4 (1.8–8.5) | 4.3 (1.8–8.3) | 4.5 (1.9–8.3) |
| Prior COPD admission in last 3 years | 346 (1.2%) | 352 (1.2%) | 365 (1.3%) | 366 (1.3%) | 323 (1.2%) | 285 (1.2%) |
| Prior cancer admission in last 3 years | 1278 (4.3%) | 1332 (4.4%) | 1266 (4.5%) | 1212 (4.2%) | 1122 (4.2%) | 1044 (4.4%) |
| Follow-up duration (months) for HES outcomes (months), median (IQR) | 47 (36–47) | 47 (36–47) | 45 (36–47) | 46 (33–47) | 45 (35–47) | 45 (33–47) |

CKD, chronic kidney disease; COPD, chronic obstructive pulmonary disease; CVD, cardiovascular disease; GFR, glomerular filtration rate; HES, Hospital Episode Statistics.

**Table 2** Practice characteristics by practice coding sextile

| Practice coding sextile | S1 | S2 | S3 | S4 | S5 | S6 |
|---|---|---|---|---|---|---|
| Practice percent coded CKD | <54.8% | 54.8%–65.5% | 65.5%–73.9% | 73.9%–80.7% | 80.7%–87.5% | ≥87.5% |
| No of practices | n=106 | n=106 | n=106 | n=107 | n=106 | n=106 |
| Practice percent coded CKD, mean (SD) | 44.2% (9.8%) | 61.2% (3.5%) | 70.0% (2.4%) | 77.5% (2.0%) | 84.1% (2.2%) | 91.7% (3.1%) |
| **Practice characteristics reflecting practice risk profile** | | | | | | |
| Mean adult age, mean (SD)[M1,M2,CP] | 48.9 (4.1) | 49.5 (4.7) | 49.1 (3.2) | 49.2 (3.9) | 49.5 (3.8) | 49.4 (3.4) |
| Percent male, mean (SD)[M1,M2,CP] | 49.3% (1.9%) | 48.6% (2.6%) | 49.3% (1.9%) | 49.1% (2.0%) | 49.3% (2.0%) | 49.1% (2.5%) |
| Practice median rank of IMD, median (IQR)[M1,M2,CP*] | 18236 (9574, 23,480) | 17887 (11,835, 22,921) | 17363 (9987, 24,460) | 17114 (10,392, 21,711) | 15793 (10,946, 21,345) | 16555 (12,742, 22,962) |
| Diabetes prevalence, mean (SD)†[M1,M2,CP] | 5.6% (1.3%) | 5.9% (1.4%) | 6.0% (1.5%) | 6.0% (1.5%) | 6.0% (1.3%) | 6.3% (1.5%) |
| Hypertension prevalence, mean (SD)†[M1,M2,CP] | 16.1% (4.1%) | 16.8% (3.9%) | 16.6% (3.7%) | 17.1% (4.1%) | 17.6% (3.7%) | 18.7% (4.2%) |
| CVD prevalence, mean (SD)†[M1,M2,CP] | 5.5% (1.8%) | 5.9% (2.4%) | 5.8% (1.6%) | 6.0% (1.8%) | 6.2% (1.6%) | 6.4% (1.8%) |
| **Practice testing behaviours** | | | | | | |
| Percent GFR test in last year in diabetes, mean (SD)[M2,CP] | 86.9% (12.8%) | 89.8% (5.3%) | 89.7% (4.7%) | 89.7% (5.2%) | 89.7% (10.4%) | 90.3% (5.0%) |
| Median months since last GFR test in diabetes, median (IQR) | 4.6 (3.7, 5.4) | 4.2 (3.5, 5.0) | 4.2 (3.7, 5.1) | 4.4 (3.7, 5.1) | 3.8 (3.3, 4.7) | 4.4 (3.8, 5.1) |
| Percent GFR test in last year in CKD, mean (SD)[M2,CP] | 83.3% (12.7%) | 86.6% (4.9%) | 87.0% (4.7%) | 87.2% (5.6%) | 86.6% (10.6%) | 88.3% (5.5%) |
| Median months since last GFR test in CKD, median (IQR) | 4.8 (4.1, 5.7) | 4.6 (4.1, 5.2) | 4.7 (4.0, 5.2) | 4.5 (3.8, 5.3) | 4.2 (3.6, 5.0) | 4.4 (3.9, 5.2) |
| Biochemical CKD prevalence, mean (SD)[M2,CP] | 4.2% (1.9%) | 4.0% (1.5%) | 3.9% (1.4%) | 3.8% (1.5%) | 4.0% (1.4%) | 4.0% (1.5%) |
| **Practice characteristics of the detected CKD population** | | | | | | |
| Percent CKD stages 3b-5, mean (SD)[M2,CP] | 37.0% (5.4%) | 37.5% (4.9%) | 37.5% (4.8%) | 38.3% (5.2%) | 38.0% (4.3%) | 38.0% (5.0%) |
| Percent recent COPD admission, mean (SD)[M2,CP] | 1.2% (1.0%) | 1.2% (0.7%) | 1.4% (0.9%) | 1.3% (1.0%) | 1.3% (1.1%) | 1.2% (1.0%) |
| Percent recent cancer admission, mean (SD)[M2,CP] | 4.3% (1.7%) | 4.4% (1.9%) | 4.4% (1.8%) | 4.2% (1.8%) | 4.1% (1.8%) | 4.4% (2.0%) |
| **Practice behaviours that may improve CKD outcomes** | | | | | | |
| Percent usage of ACE/ARBs in hypertension, mean (SD)[CP] | 76.0% (6.2%) | 75.5% (6.1%) | 77.1% (5.3%) | 75.9% (5.9%) | 76.5% (4.6%) | 78.2% (5.5%) |
| Percent usage of statins in diabetes, mean (SD)[CP] | 82.2% (5.8%) | 83.2% (5.8%) | 82.9% (5.5%) | 84.2% (5.8%) | 84.8% (5.3%) | 85.0% (5.5%) |
| Percent usage of statins in CVD, mean (SD)[CP] | 91.4% (3.4%) | 92.2% (3.5%) | 92.8% (2.9%) | 93.0% (2.6%) | 93.1% (2.6%) | 93.0% (3.5%) |
| Percent meeting blood pressure target in last year in CKD, mean (SD)[CP] | 56.8% (7.4%) | 57.0% (6.2%) | 57.9% (7.0%) | 56.8% (7.4%) | 58.3% (7.4%) | 60.2% (6.7%) |
| Percent ACR/PCR test in last year in CKD, mean (SD)[CP] | 40.1% (13.3%) | 49.1% (13.4%) | 55.9% (11.9%) | 61.3% (13.3%) | 62.5% (17.0%) | 68.9% (15.7%) |
| Percent influenza vaccination in last year in CKD, mean (SD)[CP] | 73.3% (15.4%) | 77.0% (7.4%) | 77.6% (5.7%) | 78.8% (5.6%) | 79.3% (9.5%) | 80.5% (6.3%) |
| Percent pneumococcus vaccination in past 5 years in CKD stages 4–5, mean (SD)[CP] | 15.3% (14.9%) | 18.2% (16.8%) | 15.2% (14.5%) | 19.1% (17.0%) | 17.1% (15.7%) | 18.2% (18.6%) |

M1 indicates variables included in statistical analysis model 1 (primary analysis).
M2 indicates variables included in statistical analysis model 2 (secondary analysis).
CP indicates variables included in statistical analysis of practice behaviour variables.
*IMD is ranked for all lower super output areas in England (1=worst, 32 844=best); summary statistics are based on IMD rankings of patients extracted from practices only and should be interpreted carefully.
†Prevalence statistics use adult population as denominator.
ACEi, angiotensin-converting enzyme inhibitors; ACR, albumin to creatinine ratio; ARB, angiotensin-II receptor blockers; CKD, chronic kidney disease; COPD, chronic obstructive pulmonary disease; CVD, cardiovascular disease; GFR, glomerular filtration rate; IMD, Index of Multiple Deprivation; PCR, protein to creatinine ratio.

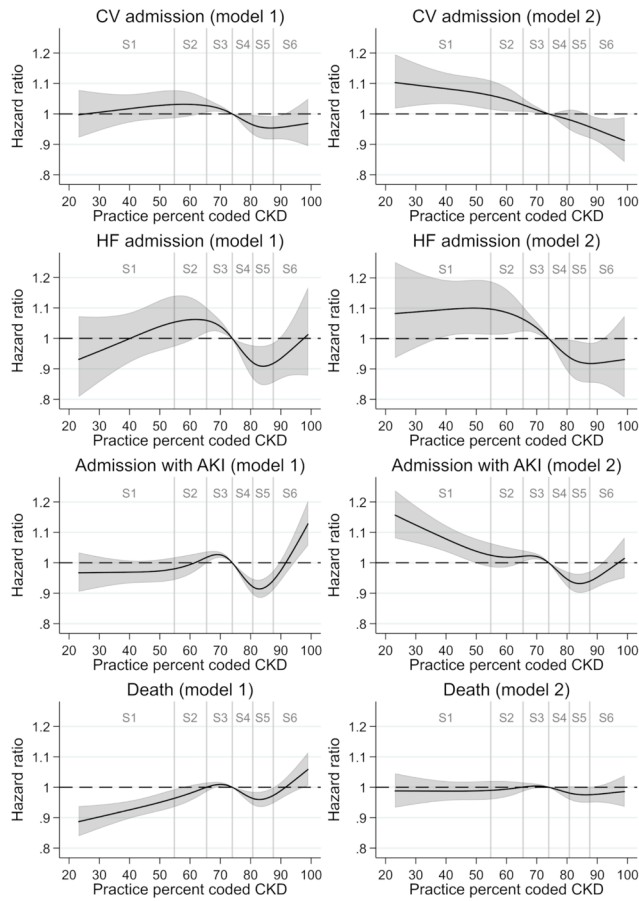

**Figure 2** HR curves for all outcomes according to practice percent coded CKD. Analysis includes all patients with confirmed CKD. Primary exposure is (continuous) practice percent coded CKD, with median practice coding (74% coded) as the reference group. Model 1 adjusts for practice risk profile characteristics: mean age, percent male, median rank of index of multiple deprivation, diabetes prevalence, hypertension prevalence, CVD prevalence. Model 2 adjusts for model 1 variables and additionally for practice characteristics of CKD population and testing behaviours: percent of CKD cases at stages 3b–5, percent of patients with CKD admitted for COPD in last 3 years, percent of patients with CKD admitted for cancer in last 3 years, percent GFR test in last year in diabetes, percent GFR test in last year in CKD, percent of adult population with CKD. Labels S1–S6 descriptively indicate sextiles of practice percent coded CKD, with each sextile representing one sixth of all practices. GFR, glomerular filtration rate; CKD, chronic kidney disease; COPD, chronic obstructive pulmonary disease; CVD, cardiovascular disease.

those with diabetes (78.6%) than those without diabetes (71.4%). Coding performance was more variable in early stage CKD and non-diabetic CKD (online supplemental figure 1).

Practice characteristics were generally well balanced in coding sextiles 2–5 (table 2). At the extremes (sextiles 1 and 6), higher coding practices had on average higher prevalence of coded comorbidities, were more deprived, and performed more regular and complete GFR testing

in those at risk. Practice behaviours that may improve CKD outcomes showed trends of improved performance in higher coding practices, across the spectrum of CKD coding. In particular, ACR/PCR testing was substantially higher in higher coding practices, and influenza vaccination rates were also markedly higher.

## Outcomes

Of 167 208 patients with CKD identified from the NCKDA database, after national linkage we found that 563 deaths had occurred but had not been reported on primary care systems at date of data extraction, leaving 166 645 eligible for outcomes analysis. Median follow-up duration was 3.8 years (range 1 day to 3.9 years) for HES outcomes and 4.3 years (range 1 day to 4.4 years) for mortality, with no meaningful differences in follow-up between sextiles. Crude event rates by sextile are shown in online supplemental table 3,4 and online supplemental figure 2,3.

## Main adjusted Cox regression analyses

Figure 2 demonstrates how individual patient risks of the four studied outcomes differ according to practice CKD coding, compared with a patient in an averagely performing practice. In model 1 analyses, inverted S-shaped HR curves suggest that confounding at the extremes of practice coding may distort the association between practice CKD coding and adverse outcomes. After further adjustments (model 2), curves become flatter (approaching linearity). Wide CIs at lower levels of CKD coding reflect poor precision in HR estimates due to sparse data. (Crude analyses and additional sequential adjustments in the CKD population and subgroups (detailed in online supplemental information 2) are shown in online supplemental figures 4–23).

There are strong trends, particularly among sextiles 2–5 (55%–88% coded), of reduced rates of outcomes with improved practice CKD coding in fully adjusted analyses (model 2). Compared with patients in the averagely performing practice (74% coded), patients in practices coding only 55% of CKD cases had a significantly higher rate of CV hospitalisations (HR 1.061 (95% CI 1.015 to 1.109)) and HF hospitalisations (HR 1.097 (95% CI 1.013 to 1.187)). Patients in practices coding 88% of CKD cases had a significantly reduced rate of CV hospitalisations (HR 0.957 (95% CI 0.920 to 0.996)) and HF hospitalisations (HR 0.918 (95% CI 0.855 to 0.985)), compared with the averagely performing practice. The percentage of preventable events over a period of 3.8 years (median follow-up time) for an improvement in practice coding from 55% to 88% of CKD cases (attributable fraction) was 9.0% for first CV hospitalisations and 16.0% for first HF hospitalisations, under assumption of causality after Cox modelling (model 2). Trends of a small reduction in AKI but no substantial change in rate of deaths were also observed as CKD coding increased. (Additional results of analysis of a single linear practice coding term (where appropriate) are shown in online supplemental table 5 and online supplemental figure 24).

**Table 3** Adjusted HRs for the association between practice behaviour variables and CV events, sorted by point estimate

| Practice behaviour | HR (95% CI) |
|---|---|
| Percent usage of ACEi/ARBs in hypertension (>76.6%) | 0.956 (0.929 to 0.983)* |
| Percent ACR/PCR test in last year in CKD (>58.7%) | 0.968 (0.939 to 0.998)* |
| Percent usage of statins in CVD (>93.0%) | 0.972 (0.942 to 1.003) |
| Percent pneumococcus vaccination in past 5 years in CKD stages 4–5 (>12.5%) | 0.982 (0.955 to 1.010) |
| Percent meeting blood pressure target in last year in CKD (>57.8%) | 0.992 (0.963 to 1.028) |
| Percent usage of statins in diabetes (>84.1%) | 0.995 (0.965 to 1.026) |
| Percent influenza vaccination in last year in CKD (>78.8%) | 0.998 (0.968 to 1.028) |

Analysis adjusted for practice characteristics: mean age, percent male, median rank of IMD, diabetes prevalence, hypertension prevalence, CVD prevalence, percent of CKD cases at stages 3b–5, percent of patients with CKD admitted for COPD in last 3 years, percent of patients with CKD admitted for cancer in last 3 years, percent GFR test in last year in diabetes, percent GFR test in last year in CKD, percent of adult population with CKD.
*95% confidence interval excludes 1
ACEi, ACE inhibitors; ACR, albumin to creatinine ratio; ARB, angiotensin-II receptor blocker; CKD, chronic kidney disease; CV, cardiovascular; CVD, cardiovascular disease; PCR, protein to creatinine ratio.

Subgroup analyses showed a steeper reduction in CV hospitalisations and HF hospitalisations with increasing practice coding among CKD stages 3b–5 than in CKD stage 3 both in spline regression analyses and those assuming linear effects of practice CKD coding (online supplemental figures 5,6,10,11,24).

### Practice behaviours analyses

Analysis of the association between practice behaviours and CV hospitalisations adjusted for all confounders showed a significant reduction in rate of CV hospitalisations for patients with CKD belonging to practices with greater than average usage (median 76.6%) of ACEi/ARBs in hypertension compared with practices with lower than average usage (HR 0.956 (95% CI 0.929 to 0.983)) (table 3). Practice ACR/PCR testing in CKD was also associated with a reduction in rates of CV hospitalisations in CKD (HR 0.968 (95% CI 0.939 to 0.998)). Results for analyses of AKI, HF and deaths are available in online supplemental tables 6–8. In brief, usage of ACEi/ARBs was the most consistently dominant predictor, being strongly associated with a reduction in events across all outcomes. This was followed by ACR/PCR testing, which was associated with a reduction in all outcomes except deaths, and usage of statins in CVD, which was associated with a reduction in rate of CV and AKI events.

### DISCUSSION

Higher levels of practice CKD coding were associated with lower rates of hospitalisation for CV and HF events among patients with confirmed CKD, after adjusting for practice characteristics. Reductions in hospitalisation rates were strongest in CKD stages 3b-5, although greatest opportunities for improvement are in CKD stage 3a where practice variation in coding was much wider. There was no difference in death rates according to practice CKD coding (although the relationship was less clear among CKD stages 3b-5). Findings were limited by duration of follow-up with longer-term benefits of CKD coding not yet apparent. Practice behaviours associated with CKD coding including usage of ACEi/ARB therapy and ACR/

PCR testing were independent predictors of reduction in hospitalisation rates.

There are very limited studies looking at the impact of recognition and diagnostic coding of CKD in primary care as many health systems use coded disease as opposed to laboratory records to identify patients. It is possible that some patients with 2 eGFR measures <60 mL/min/1.73 m$^2$ more than 3 months apart are not recognised by GPs as having CKD because of concerns of overdiagnosis, for example, in elderly patients without hypertension or in patients recovering from AKI, despite these patients meeting the accepted definition of CKD. Furthermore, numerous studies have demonstrated disparities in CKD coding efforts with younger patients, those from deprived backgrounds, and ethnic minorities being less commonly coded than their counterparts,[7 15] leading to concerns around equity of care. Recent studies have shown an association between CKD coding and interventions known to reduce CV risk such as prescription of statins and antihypertensive agents,[15 16] and CKD coding may play a role in triggering further long-term treatment efforts with potential to reduce patient risks. Our study identified a reduced burden of CV and HF hospitalisations for practices coding more CKD, and a reduced burden of hospitalisations for practices providing more interventions (associated with CKD coding) that are likely to improve CKD outcomes.

A key strength of this study is the large sample size, including data from 167 208 patients with CKD. Data were extracted from GP practices in England with a similar age–sex distribution to the whole population, so findings are likely to be generalisable to the wider population. By studying the association between practice-level CKD coding and patient-level outcomes, we were able to eliminate a lot of confounding due to individual patient characteristics that would be present in a conventional study design using patient-level exposure with unmeasured confounders. This was demonstrated by the balanced risk profile in patient characteristics observed across sextiles (which is likely to extend also to unmeasured characteristics). This is a major benefit over original analyses of the NCKDA[12] (online supplemental figure 25), which

showed a very strong association between individual patient CKD coding and risk of outcomes (CV events, AKI, death) but with a high risk of confounding due to coding efforts being associated with perceived patient risk, and some potentially important risk factors missing from the database.

A potential weakness is that included practices had volunteered to participate in an audit of care. CKD coding may have been higher in recruited practices than the general population which may have impacted on estimated strengths of associations; benefits of coding in the wider practice population may be larger than estimated. Risk factor evaluation mostly relied on comorbidity coding and it is not clear whether small differences in risk factor prevalence reflected true morbidity or GP behaviour. Assessment of eligibility of patients for analysis also relied on availability of repeat creatinine tests over time, which may depend on patient risk, and earlier stage CKD cases or more severe cases managed solely in secondary care may be disproportionately missing. Nevertheless, this identified CKD population may stand to benefit most imminently from improvements in primary care, assuming GPs target further coding efforts to patients already identified as at risk and with creatinine test results compatible with CKD. While there was a small signal of more frequent creatinine testing with increasing practice CKD coding, this was only at the extremes, and distribution of CKD severity appeared generally very well balanced across practice coding sextiles. Practice characteristics were analysed differently depending on whether they were likely to confound analyses or lie on the causal pathway, however, we could not verify if our assumptions were reasonable, and misspecification could affect reliability of conclusions. For example, practice management of hypertension with ACEi/ARB therapy may plausibly confound analyses (if hypertension management and CKD coding share a common cause, such as practice funding or clinical expertise) or lie on the causal pathway (if management of hypertension occurs as a consequence of CKD coding). Our findings for AKI are likely affected by outcome misclassification as hospital codes were used to detect AKI events, which may have led to underestimation of the number of events and lack of power to detect an association. We did not have enough dialysis events to allow evaluation of the impact of practice coding on outcomes. These data precede the use of SLGT2-inhibitor drug treatment in HF and albuminuric kidney disease in UK primary care.

## Conclusions

Rates of CV and HF events were lower for patients belonging to practices coding more CKD, supporting the argument that CKD coding in primary care may contribute to improvement in patient outcomes. While the presence of unmeasured confounding cannot be ruled out, this is in agreement with other studies conducted in this setting.[15–17] High-quality evidence supporting our findings is available from clinical trials and systematic reviews which underline the benefits of use of interventions in early-stage CKD, including ACEi/ARB therapy to control hypertension and statin therapy to reduce CV risk.[18–20] This study suggests that reductions in key adverse events for patients with CKD could be made by improvements to GP practice identification and coding of CKD as these are associated with subsequent care efforts that are known to prevent poor outcomes.

**Author affiliations**
[1]Department of Non-Communicable Disease Epidemiology, London School of Hygiene and Tropical Medicine, London, UK
[2]Cardiovascular Epidemiology Unit, University of Cambridge, Cambridge, UK
[3]Department of Renal Medicine, University College London, London, UK
[4]UK Renal Registry, UK Kidney Association, Bristol, UK
[5]Department of Renal Medicine, Royal Derby Hospital, Derby, UK
[6]Department of Cardiovascular Sciences and NIHR Leicester Biomedical Research Centre, University of Leicester, Leicester, UK
[7]Institute of Health Informatics, University College London, London, UK
[8]British Heart Foundation Data Science Centre, London, UK
[9]No affiliation, retired, York, UK
[10]Director of Policy, Kidney Care UK, Alton, UK
[11]Wolfson Institute of Population Health, Queen Mary University of London, London, UK

**Acknowledgements** We thank all the patients who have provided their data to the audit analyses, and patient organisations, especially KidneyCare UK and the National Kidney Foundation, and those involved in the set up and conduct of the National Chronic Kidney Disease Audit, which provided a large database of patients with CKD, without which this research would not have been possible. This includes (apart from the listed authors): Matthew Harker, Yvonne Silove, Tasneem Hoosain, Nick Wilson, Ronnie Moodley, Chris Gush, Liam Smeeth, Ron Cullen, Fergus Caskey, Andy Syme, Richard Gunn, Paul Wright, Hugh Gallagher, Sion Edwards, Nick Palmer, Anita Sharma, Kate Cheema, Kathleen Mudie.

**Contributors** DN, FC, BC, SH, LK and DP-M planned the analysis. FC conducted analysis, interpreted results and reported the work. DN, LK, DP-M, DW, SH, RS, RF, DA, SD, KG, FL and BC contributed insights in the interpretation of results, provided updates to manuscript text, and read and approved the final manuscript. FC is the guarantor and accepts full responsibility for the finished work and conduct of the study, had access to the data, and controlled the decision to publish.

**Funding** This research was funded by the UK MRC, grant reference MR/N013638/1. The funding body had no role in the design of the study, collection, analysis, or interpretation of data and no role in writing of the manuscript.

**Competing interests** David C Wheeler has an ongoing consultancy contract with AstraZeneca and has received honoraria, consultancy fees or speaker fees from Amgen, Astellas, Bayer, Boehringer Ingelheim, GlaxoSmithKline, Gilead, Janssen, Napp/Mundipharma, Merck Sharp and Dohme, Tricida, Vifor and Zydus. David Adlam has received research funding from Abbott vascular to support a clinical research fellow; he has also received funding from AstraZeneca inc. for unrelated research and has undertaken consultancy for General Electric inc. to support research funds. Dorothea Nitsch reports grants unrelated to this work from the National Institute for Health Research, Medical Research Council (MRC), the Health Foundation and GlaxoSmithKline.

**Patient and public involvement** Patients and/or the public were involved in the design, or conduct, or reporting, or dissemination plans of this research. Refer to the Methods section for further details.

**Patient consent for publication** Not applicable.

**Ethics approval** The NCKDA research database has national ethical approval (REC reference: 17/LO/0049), section 251 approval (CAG reference number: 17CAG0058). NCKDA steering group approval and LSHTM ethics approval (reference: 21689) were granted for this analysis.

**Provenance and peer review** Not commissioned; externally peer reviewed.

**Data availability statement** No data are available. The data that support the findings of this study are stored at University College London (UCL) but restrictions apply to the availability of these data, which were used under license for the current study, and so are not publicly available.

**ORCID iDs**
Faye Cleary http://orcid.org/0000-0001-8401-2305
Sally Hull http://orcid.org/0000-0002-8691-7519
Ben Caplin http://orcid.org/0000-0001-9544-164X

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
