## [Reviewer comments · BMJ Open]

ARTICLE DETAILS

TITLE (PROVISIONAL)	Association between practice coding of chronic kidney disease (CKD) in primary care and subsequent hospitalisations and death: a cohort analysis using national audit data
AUTHORS	Cleary, Faye; Kim, Lois; Prieto-Merino, David; Wheeler, David; Steenkamp, Retha; Fluck, Richard; Adlam, David; Denaxas, S; Griffith, Kathryn; Loud, Fiona; Hull, Sally; Caplin, Ben; Nitsch, Dorothea

VERSION 1 – REVIEW

REVIEWER	Stengel, Benedicte Centre de recherche en Epidemiologie et Sante des Populations
REVIEW RETURNED	06-Jun-2022

GENERAL COMMENTS	Thank you for giving me the opportunity to review this manuscript. This is an original study investigating the impact of CKD coding by GP practices on adverse outcomes in patients with eGFR-based CKD. This study has several strengths including the analysis of associations between practice-level (rather than patient-level) CKD coding and patient-level outcomes, and the availability of detailed patient- and practice-level data. Statistical analyses are appropriate and comprehensive. Major comments concern potential information bias, discussion of findings and clinical implications. Major comments 1. To be eligible for this study, patients were required to have 2 eGFR <60 separated by at least 90 days, with no time limit beyond. Moreover, it is unclear, on top of page 8, whether study follow-up started after the first or second measurement. Clarifying this point is important as well as discussing potential selection or information (immortal) bias resulting from the choice of one or the other method.2. The number of events for each studied outcomes should be provided in addition to event rates in Supplementary Table 3. Page 17, it is written that "563 eligible patients died before start of follow-up". Does this mean that patients could be eligible for the study even if they had died before starting follow-up?3. According to the author hypotheses, this study shows reduced risks of hospitalizations for CVD and HF associated with practice CKD coding, but the association with AKI is less convincing. Could that be due to the use of diagnosis codes to ascertain AKI which may have underestimated the number of events? Further discussion about AKI findings is expected.4. A limitation of the study is that it does not assess the association of practice CKD coding with progression to kidney failure with (or without) replacement therapy (KFRT). This would
--

	be the main outcome for which improvement may be expected from CKD recognition. Would this outcome be available for analysis? If not, this may be discussed as a study limitation and/or a possible perspective. 5. In this analysis, CKD coding by GP is used as a proxy for CKD recognition. Nevertheless, it is possible that patients with 2 eGFR < 60 are not recognized as having CKD because GPs have additional information which does not support CKD diagnosis, e.g., in elderly women without hypertension and no sign of kidney damage, or in patients slowly recovering from AKI. The higher variability of “coding performance” in early CKD stage may not only reflect lack of CKD recognition, but also willingness to avoid overdiagnosis. This point could be discussed. 6. Discussing the potential impact of CKD coding on COVID outcomes seems out of scope, while several study findings are not. In summary, this study shows that CKD coding performance used as a proxy for CKD recognition is independently associated with CV outcomes. However, it is not associated with mortality, and the association with key adverse kidney outcomes is uncertain for AKI, and not addressed for KFRT. Clinical implications of these findings as a whole and research perspective may be further discussed. Minor The abstract should report findings for mortality and AKI outcomes. Several acronyms are not developed in the text or in the Tables (missing legend) Page 8 : provide units for ACR and PCR. Check ACR ≥ 700. Should be ≥ 70?
--	--

REVIEWER	Hunter, Barbara The University of Melbourne, Department of General Practice
REVIEW RETURNED	18-Jul-2022

GENERAL COMMENTS	Interesting paper on important issue.
---------------------------------------

VERSION 1 – AUTHOR RESPONSE

Reviewer 1	Thank you for giving me the opportunity to review this manuscript. This is an original study investigating the impact of CKD coding by GP practices on adverse outcomes in patients with eGFR-based CKD. This study has several strengths including the analysis of associations between practice-level (rather than patient-level) CKD coding and patient-level outcomes, and the availability of detailed patient- and practice-	Thank you for your constructive and detailed feedback.
------------	--	--

	level data. Statistical analyses are appropriate and comprehensive. Major comments concern potential information bias, discussion of findings and clinical implications. Major comments		
1	To be eligible for this study, patients were required to have 2 eGFR <60 separated by at least 90 days, with no time limit beyond. Moreover, it is unclear, on top of page 8, whether study follow-up started after the first or second measurement. Clarifying this point is important as well as discussing potential selection or information (immortal) bias resulting from the choice of one or the other method.	Thank you. We restricted our analyses to people who at practice data extraction date (start of follow-up time) had data compatible with CKD on their practice file. That means, on the date of practice data extraction (start of follow-up, as per existing text on page 8) we looked back at the existing health records and identified all people who had 2 eGFR values <60 ml/min/1.73m² more than 3 months apart. That means by default we only look at people with more than 2 measurements after having survived up to the date that the audit data extraction was done to quantify the quality of care. To make it clearer we have added text on page 6 to say “In brief, the audit is a snapshot of care at the time point of audit data extraction.”	Page 6.
2	The number of events for each studied outcomes should be provided in addition to event rates in Supplementary Table 3. Page 17, it is written that “563 eligible patients died before start of follow-up”. Does this mean that patients could be eligible for the study even if they had died before starting follow-up?	We have updated Supplementary Table 3 which reports on recurrent events and provided a new table for first events (new Supplementary Table 4). These two tables correspond to supplementary figures 2 and 3. The number of events on which event rates are based are now included in both tables. 563 patients who appeared to be alive and managed by the GP practice at NCKDA data extraction and hence eligible for analysis were later found to have in fact died before start of follow-up (date of NCKDA data extraction) after inspection of the ONS mortality data after data linkage. This is due to delayed reporting of deaths to the primary care provider. We have rephrased this to “Of 167,208 CKD patients identified from the NCKDA database, after national linkage we found that 563 deaths had occurred but had not been reported on primary care	Page 17

		systems at date of data extraction, leaving 166,645 eligible for outcomes analysis.”	
3	According to the author hypotheses, this study shows reduced risks of hospitalizations for CVD and HF associated with practice CKD coding, but the association with AKI is less convincing. Could that be due to the use of diagnosis codes to ascertain AKI which may have underestimated the number of events? Further discussion about AKI findings is expected.	Thank you for raising this point. Yes, AKI coding may well be misclassified. We have added this to the discussion. “Our findings for AKI are likely affected by outcome misclassification as hospital codes were used to detect AKI events which may have led to underestimation of the number of events and lack of power to detect an association.”	Page 21
4	A limitation of the study is that it does not assess the association of practice CKD coding with progression to kidney failure with (or without) replacement therapy (KFRT). This would be the main outcome for which improvement may be expected from CKD recognition. Would this outcome be available for analysis? If not, this may be discussed as a study limitation and/or a possible perspective.	We do not have enough dialysis events to allow evaluation of the impact of practice coding on outcomes. (Added to limitations)	Page 21
5	In this analysis, CKD coding by GP is used as a proxy for CKD recognition. Nevertheless, it is possible that patients with 2 eGFR < 60 are not recognized as having CKD because GPs have additional information which does not support CKD diagnosis, e.g., in elderly women without hypertension and no sign of kidney damage, or in patients slowly recovering from AKI. The higher variability of “coding performance” in early CKD stage may not only reflect lack of CKD recognition, but also willingness to avoid overdiagnosis. This point could be discussed.	We have added this point to the discussion on page 22. “It is possible that some patients with 2 eGFR < 60 ml/min/1.73m ² more than 3 months apart are not recognized by GPs as having CKD because of concerns of overdiagnosis, for example in elderly patients without hypertension or in patients recovering from AKI, despite these patients meeting the accepted definition of CKD. Furthermore, numerous studies have demonstrated disparities in CKD coding efforts with younger patients, those from deprived backgrounds, and ethnic minorities being less commonly coded than their counterparts,[7,16] leading to concerns around equity of care. Recent studies have shown an association between CKD coding and interventions known to reduce CV risk such as prescription of statins and anti-hypertensive agents,[16,17] and CKD coding may play a role in triggering further long-term treatment efforts with potential to reduce patient risks. Our study identified a reduced burden of CV	Page 20

		and HF hospitalisations for practices coding more CKD, and a reduced burden of hospitalisations for practices providing more interventions (associated with CKD coding) that are likely to improve CKD outcomes.”	
6	Discussing the potential impact of CKD coding on COVID outcomes seems out of scope, while several study findings are not. In summary, this study shows that CKD coding performance used as a proxy for CKD recognition is independently associated with CV outcomes. However, it is not associated with mortality, and the association with key adverse kidney outcomes is uncertain for AKI, and not addressed for KFRT. Clinical implications of these findings as a whole and research perspective may be further discussed.	We removed the text on COVID as requested.	Page 20
Minor	The abstract should report findings for mortality and AKI outcomes. Several acronyms are not developed in the text or in the Tables (missing legend) Page 8 : provide units for ACR and PCR. Check ACR \geq 700. Should be \geq 70?	Added to abstract as requested. We checked acryonyms and spelt them out when not developed in the text (e.g. Sextile and LSOA). Thank you for spotting the typo – it is \geq 70, now amended. Units are also added.	Abstract Table 2 Page 8
Reviewer 2			
	Comments to the Author: Interesting paper on important issue.	Thank you for your supportive feedback.	

1

VERSION 2 – REVIEW

REVIEWER	Stengel, Benedicte Centre de recherche en Epidemiologie et Sante des Populations
REVIEW RETURNED	01-Sep-2022

GENERAL COMMENTS	Authors appropriately addressed my comments with a minor exception. The number of events for each studied outcomes were not included in revised Supplementary Tables 3 and 4 as requested, although it is mentioned that this had been done .
--